

**Impact of naval traffic on the sediment transport of the Port of Genoa – a modelling study**
*Antonio Guarnieri [1], Sina Saremi [2], Andrea Pedroncini [3] Jacob H. Jensen [2], Silvia Torretta [3] Marco Vaccari [4],*
*Caterina Vincenzi [4]*
*(1) Istituto Nazionale di Geofisica e Vulcanologia, Sezione di Bologna, Via D. Creti, 12, 40128 Bologna, Italy*
*(2) DHI, Horsholm, Denmark*
*(3) DHI S.r.l., Via Bombrini 11/12, 16149 Genova*
*(4) Autorità di Sistema Portuale del Mar Ligure Occidentale (Genova), Palazzo San Giorgio - Via della Mercanzia 2*
*Corresponding author: Antonio Guarnieri; antonio.guarnieri@ingv.it*



**Abstract**
The action of propellers induced jets on the seabed of ports can be responsible of erosion and deposition of sediment
around the port basin, potentially inducing important variations of the bottom topography in the medium to long time
scales. Such dynamics constantly repeated for long periods can result in drastic reduction of ships' clearance - in the
case of accretion - or might be a threat for the stability and duration of the structures - in the case of erosion. These
sediment related processes are sources of problems for the port managing authorities, both for the safety of navigation
and for the optimization of the management and maintenance activities of the ports' bottom and infrastructures.
In the present work we study the erosion and sediment transport induced by the action of the vessel propellers of
naval traffic in the passenger Port of Genoa (Italy) by means of integrated numerical modeling and we propose a
novel methodology and state of the art modeling science-based tools useful to optimize and efficiently plan the ports
managing activities and the of maintenance of ports seabed.
**1 - Introduction**
Operational activities of harbors and ports are tightly related to the local bathymetry, which must be as deep as to
guarantee the regular passage, maneuvering and berthing of ships. On the contrary, ships clearance is often so limited
that the safety of in-port navigation might be at risk and ships may even hit the sea bed in extreme cases. This is a
source of high criticalities, not only for safety sake, but also for the consequent rise of problems related to an efficient
management and maintenance of the bottom and of the port infrastructure in general.
Ships' traffic inside ports is responsible for the generation of intense current jets produced by the action of the main
propellers, as sketched in Figure 1. Such velocities induce shear stresses on the sea bottom which can possibly result
in sediment resuspension, when exceeding the critical stress for erosion (Van Rijn, 2007, Soulsby et al., 1994, Grant
and Madsen, 1979). Before depositing back onto the sea floor, the re-suspended sediment might be widely transported
around the basin by the combined effect of natural currents, such as those induced by tides, winds or density
gradients, and vessels' related currents, such as those directly induced by propellers or again by the movement and
displacement of the ships. Therefore, the continuous traffic in and out ports could result in the displacement of a great
amount of seabed material which can, in turn, induce important variations of the bathymetry in the medium to long
time scales. The result of these variations is the possible formation of erosional or depositional trends for specific
areas of port basins.




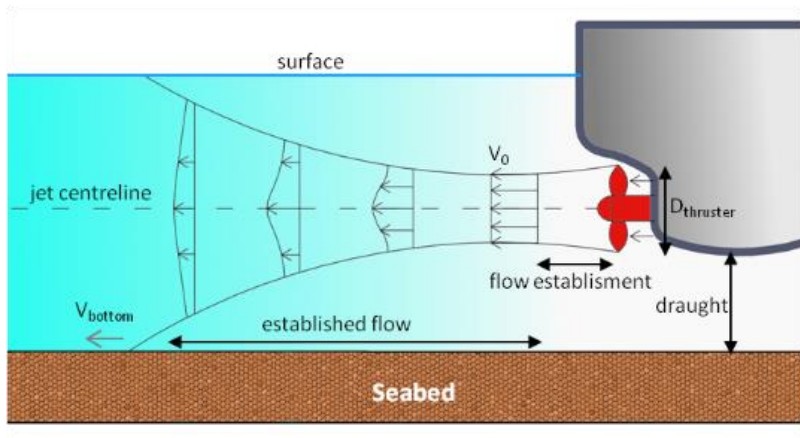


**Figure 1 -** Example of propeller induced jet of a moving ship (main propulsion without rudder)


These processes can have direct impact on the operability of ports and on safety depths for navigation (Mujal-Colilles
et al., 2016, Castells et al., 2018). If such dynamics are particularly relevant and fast (bottom accretion of the order of
tens of centimeters per year, or even higher) they induce port authorities to undergo dredging operations for the
maintenance of the seabed in order to fully recover the required clearance and operational conditions necessary for
undisturbed ships motion, maneuvering and berthing.
The majority of published literature and studies about the effects of ships' propellers on port sediments and structures
is experimental., mainly conducted in laboratories with the use of physical models (Castells et al. 2018), while port
authorities suffer from the lack of practical instruments available to provide robust and scientifically based studies and
predictions of the described processes. Such tools would allow for an aware planning of specific actions aimed at the
maintenance of the seabed. This would help to guarantee the continuity in the operational activities of ports on one
side, and to the optimization of the involved economic resources on the other side. In fact, the need of unplanned
maintenance activities usually implies additional costs due to operating in emergency conditions and in some cases to
the partial interruption of the service.
The integrated numerical modeling of hydrodynamics and sediment transport may represent an important aid to Port
Authorities and more broadly to port managers and operators. It could be used to reproduce and better understand the
seabed sediment dynamics induced by ships' propellers on the short, medium and long-time scales and so provide the
needed tools in the perspective of an efficient operational maintenance of the seabed. Such tools can be used in
delayed mode in order to reproduce the major sediment processes in the past - as it is the present case - or even in
forecast mode through the implementation of real time operational services.



So far, the issue of propeller's induced jet has been mainly studied through empirical approaches, usually relying
either on the German method (MarCorm WG, 2015, Grabe, et al. 2015, Abromeit et al., 2010,), or on the Dutch
method (CIRIA et al., 2007). In such approaches, empirical formulas are introduced in order to estimate the propeller
wash on the sea bed in terms of induced velocities and resulting induced shear stresses, depending on specific
characteristics of the ships and ports of interest, such as the propeller's typology, diameter, rotation rate and ship's
draught. The resulting induced velocities are usually considered only locally for the technical design of mooring
structures and for considerations on the protection of port's infrastructures in general. Besides the assumptions
introduced in the empirical formulas, such an approach is punctual and does not provide the full picture of the three
dimensional evolution of the induced jet throughout the water column at any distance from the propeller, in any
location of the port. The tool is therefore not suitable for a comprehensive management of the ports in a broader way.
The present work shows a pilot study of seabed evolution induced by ships' propellers in the passenger area of the
Port of Genoa (Figure 2), where the naval traffic involves mainly passenger vessels (ferries and cruise ships, generally
self-propelled) and where the resulting sediment dynamics (erosion/deposition rates) is particularly relevant:
estimated in the order of several tens of centimeters per year (direct communication from the Port Operators and
analysis of bathymetric surveys at different time). The proposed approach is based on fully integrated high resolution
numerical modeling of three-dimensional hydrodynamics and sediment transport.
The manuscript is organized as follows: in Sect. 2 we introduce the adopted methodology, while the data available for
the study are presented in Sect. 3. Sect. 4 describes the numerical models used. The results of the numerical
simulations are presented in Sect. 5 and discussed in Sect. 6, which offers some conclusions as well.

**2 – Methods**
The study is based on the latest versions of the hydrodynamic and mud transport models MIKE 3 FM (DHI, 2017)
which will be described in detail in Sect. 3 and in APPENDICES A1 and A2.
In order to resolve in a realistic way the propellers induced jet, a very high resolution was adopted in the numerical
model both in the vertical and in the horizontal: approximately 1-2 meters and 5 meters, respectively. This, together
with the use of a non-hydrostatic version of the hydrodynamic model allowed to reproduce very accurately the
processes and the main patterns of the current field generated by the ships propellers during the navigation and
maneuvering inside the port.
As shown in Figure 2, 12 docks have been included in the study (marked with orange or red lines indicating ferry or
cruise vessels, respectively). Only passenger ships were studied. The turning basins where arriving vessels undergo
maneuvers for berthing are represented in Figure 2 with the white dashed circles marked as *a* and *b*. Circle *a* refers to



vessels berthing at docks T5 to T11, while circle *b* refers to vessels to docks T1 to T3. Finally, the turning area for
vessels arriving to docks D.L., 1012 and 1003 is at the entrance of the port and is not simulated in this study since it is
out of the area of interest.
The general methodology adopted is organized in different phases, as follows:

1. *Assessment of the naval traffic during a typical year*. This was fundamental to understand the typical
   dynamics of the naval traffic in the different sectors of the port and to identify the characteristics of the ships
   that most impact hydrodynamics and sediment re-suspension from the bottom, such as the size of the ships,
   the related draught, the dimension of the propellers and their typical rotation rates. The results of the
   analysis, which will be detailed in Sect. 4.1, led also to the definition of one most representative synthetic
   vessel for each berth of the port.

2. *Implementation of a high-resolution 3D hydrodynamic model of the port of Genoa*. The numerical
   hydrodynamic model that we implemented took into account the ship routes, both entering and exiting the
   port, as analyzed within the previous vessel traffic analysis phase. As it will be detailed in Sect. 4.1, 24
   different simulations of the hydrodynamic model have been implemented, one for each dock and route
   considered (docking and undocking). The resulting 24 different scenarios have been simulated separately.
   This allowed to analyze the effect of each vessel's passage on the induced hydrodynamics in the basin. The
   single hydrodynamic contributions were then used to drive the sediment transport model. The present
   approach won't therefore consider potential simultaneous interactions amongst hydrodynamic patterns
   generated by different propellers, assuming that very close passages of different vessels are unlikely to
   happen.

3. *Implementation of a coupled sediment transport model*. Based on the available data, a numerical model of
   sediment resuspension and transport for fine-grained and cohesive material was implemented. The model
   was coupled to the hydrodynamics resulting from the 24 different vessels scenarios. As with the
   hydrodynamic component, the simulations of the sediment model were carried out separately.

4. *Gathering of the separate results and overall analysis*. The effects of the passage of the single vessels on the
   bottom sediment have been summed-up to each other in terms of erosion/deposition according to the overall
   number of passages over the one-year period of time previously analyzed. This led us to provide information
   on the resulting annual sediment dynamics.

A semi-quantitative calibration/validation of the modeling results was possible through the comparison of the seabed
evolution reproduced with the integrated modeling system and the differential bathymetric maps derived from
different surveys of the port topography at approximately one year interval.





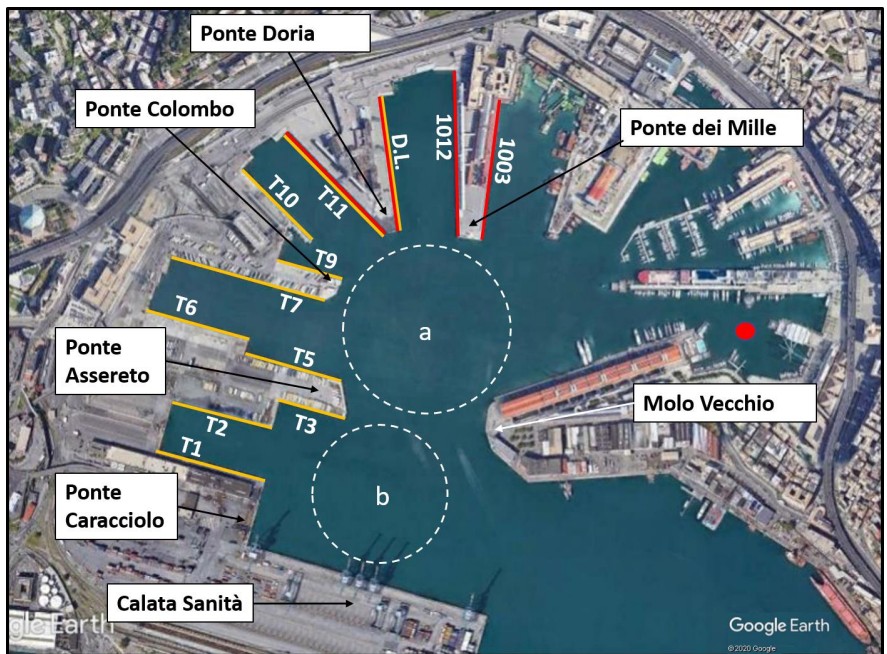


**Figure 2 - Passenger port of Genoa. The colored lines along the docks refer to the typology of the operating**


**ships: red lines indicate cruise vessels while orange lines indicate ferries. The names of the docks (in white) are**


**next to the colored lines are. The red dot represents the location of the station where sediment samples with**


**physical information on the grains are available (see Sect. 4.2). The white dashed circles marked as *a* and *b***


**represent the turning areas for vessels berthing to docks T5 to T11 and to T1 to T3**



**3 – Available data and information**
The most relevant data necessary for the implementation of the work were provided by the Port Authority of Genoa and
Stazioni Marittime SpA, which cover the role of Port Authority and main Port Operator in the target area, respectively.

**3.1 – Bathymetry**
Several bathymetry surveys of the different sectors of the port were available at different resolutions in the domain of
interest. The dataset used for the simulations was the result of the merging of the latest available surveys (March-June
2018) in the inner sectors of the port delivered on a regular grid of 5 meters of resolution. Figure 3 shows the latest
available observed bathymetry of the entire port (left panel) and a zoom focused on *Ponte Colombo* and the surrounding
basin. A few tens of meters off the right edge of *Ponte Colombo* and *Ponte Assereto* (see Figure 2) a deep natural pit in
the bathymetry is clearly visible, reaching approximately 22 meters below the water surface. This area has often been



used in the past by the Port Authority as a preferred site for dumping the sediment resulting from recurring maintenance
dredging operations of the seabed in those sectors where depositional trends are large enough to reduce vessels
clearance and to impact on the safety of navigation inside the port. Moreover, the same depressed area is largely used as
a turning area by the passenger ferries heading to docks T5, T6, T7 and T9, which cover approximately the 50% of the
entire naval traffic of the basin (see Sect. 4.1): during manoeuvres over this pit the turning ferries produce intense
turbulence which may reach the newly dumped material resulting from the dredging operations. This material is still
rather loose and consequently subject to be easily re-suspended and transported again around the port basin, nullifying
the results of the dredging operations.

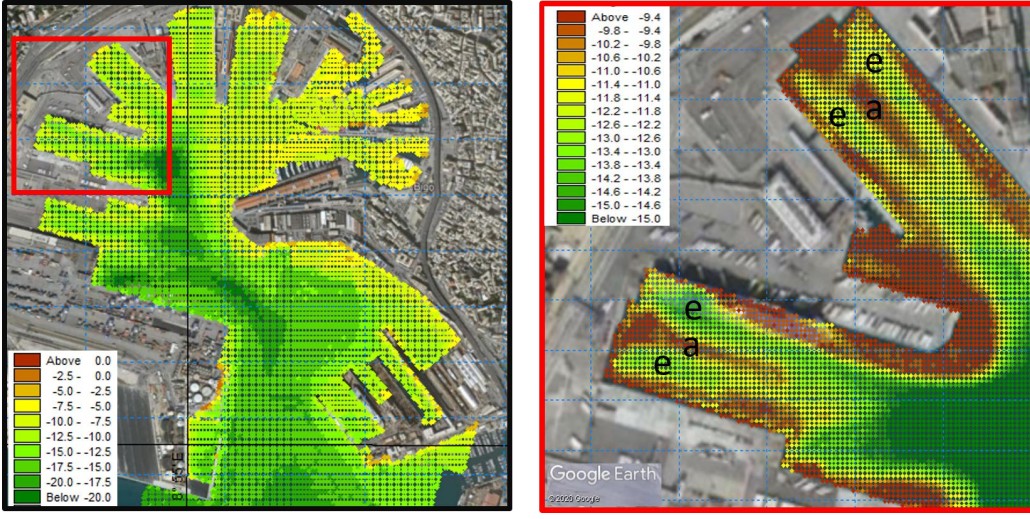


**Figure 3 - Bathymetry of the port of Genoa. Entire Passenger Port (left panel) and zoom on *Ponte Colombo* and**
**the surrounding basins (from T5 to T11, right panel)**

Additionally, the bathymetry presented in the right panel of Figure 3 shows a pattern of erosion and accumulation
common to the majority of the wet basins confined amongst the different docks. Here, the propellers activity when
vessels leave or approach the berth induces areas of erosion identified with channels of deepened bathymetry (referred
to with an "*e*" in the right panel of Figure 3, where colours are yellow-green) and areas of accumulation identified with
tongues of shallower bathymetry (referred to with an "*a*" in the right panel of Figure 3, where colours are brown).
It is important to underline that another survey covering approximately the same area as that of Figure 3 was available
for the period May-June 2017. The information resulting from the difference of such topographies, integrated with the
available information on dredging activities operated during the same period allowed to reconstruct from a semi-
quantitative point of view the sediment dynamics occurred during this time window of approximately one year. Such





information was used in the process of calibration/validation of the numerical model of sediment erosion and transport,
as detailed in Sect. 5.

**3.2 – Sediment data**
Information on the sediment textures in the sea is usually poorly available. In this case we had access to the MArine
Coastal Information sySTEm (MACISTE; http://www.apge.macisteweb.com) implemented by the Department of
Science of Earth, Environment and Life (DISTAV) of the University of Genova, where the results of several chemical
and physical sediment surveys are stored and accessible. Unfortunately, albeit the chemical information is
comprehensive, information on the grain size is rather poor for what concerns the inner area of the port. The red dot of
Figure 2 represents the only location inside the basin where the information on the texture composition and grain size
was available. These characteristics are necessary for the sediment transport model, and they were used in the
simulations for the entire domain of the numerical model (see Sect. 4.2).

**3.3 – Naval traffic**
Year 2017 was considered as a typical year from the point of view of the naval traffic in agreement with the Port
Authority of Genoa and with Stazioni Marittime SpA. The traffic was available on a daily basis and it included the
information on the docks of arrival/departure and the name of the involved vessels. The entire year was considered, in
order to account for the typical seasonality of the traffic concentration, much more relevant for passenger vessels in the
period from the end of spring to the beginning of fall.
For extra information on the characteristics of the vessels, such as length, width, tonnage, draught and typical routes
inside the port during arrivals and departures we referred to the public web page https://www.marinetraffic.com.
The outcome of the analysis will be presented in Sect. 4.1.

**4 – The numerical models**
The non-hydrostatic version of MIKE 3 HD flow model (DHI, 2017) was used to simulate the propeller induced three-
dimensional current along the port basin. The resulting hydrodynamic field was coupled with the sediment transport
module MIKE 3 MT (DHI, 2019), suitable for fine-grained and cohesive material., in order to drive the erosion,
advection-dispersion and deposition of fine sediment along the water column.

**4.1 – The hydrodynamic model**



MIKE 3 FM flow model is an ocean circulation model suitable for different applications within oceanographic, coastal
and estuarine environments at global., regional and coastal scales. It is based on the numerical solution of the Navier-
Stokes equations for an incompressible fluid in the three dimensions (momentum and continuity equations), on the
advection-diffusion of potential temperature and salinity and on the pressure equation which in the present non-
hydrostatic version is split into a hydrostatic and a non-hydrostatic component. The closure of the model is guaranteed
by the choice of a turbulence closure formulation with different possible options amongst a constant value, a
logarithmic law scheme or a k-ε scheme, which is the one used in the present implementation. The surface is free to
move and it can be solved using a sigma coordinate (as it is the case in the preset study) or a combined sigma-zed
approach. The spatial discretization of the governing equations of the model follows a cell-centered finite volume
method. In the present implementation of the model we used the barotropic density mode, thus temperature,
salinity and consequently density are constant in time and space during the simulations.
The domain of the present implementation of the model is presented in the upper panels of Figure 4. The images show
two examples of computational grids used for the simulations. In these cases the docks are T1 (left panel) and T10
(right panel) during inbound operations. The grids are a combination of unstructured triangular and quadrilateral cells
with horizontal resolution varying from 30 meters in the furthest areas from the ship trajectory, to 5 meters
approximately within the closest area to the ships propellers. The mesh is rectangular in those areas where the ships are
moving straight and the 5 meter resolution covers a corridor of approximately 50 meters of width. In the manoeuvring
areas the mesh becomes unstructured and the resolution is again 5 meters. The red lines in the middle of the 5 meter
resolution corridors of the upper panels represent the routes followed by the ships inside the port. The lower panels of
the figure are snapshots taken from the web service https://www.marinetraffic.com showing the actual routes of the
vessels birthing in the same docks as the upper panels (T1 and T10) as recorded by the AIS system mounted on the
ships. As shown in Figure 4 the reconstructed trajectories of the ships in the model are realistic and fully representative
of the real ones.
Table 1 shows the results of the traffic analysis within the Port of Genoa for year 2017 conducted on the traffic data
provided by Stazioni Marittime SpA on a daily basis. The average lengths, widths and draughts of the ships were
evaluate calculating the mean of the single quantities weighted on the number of passages occurring per year.





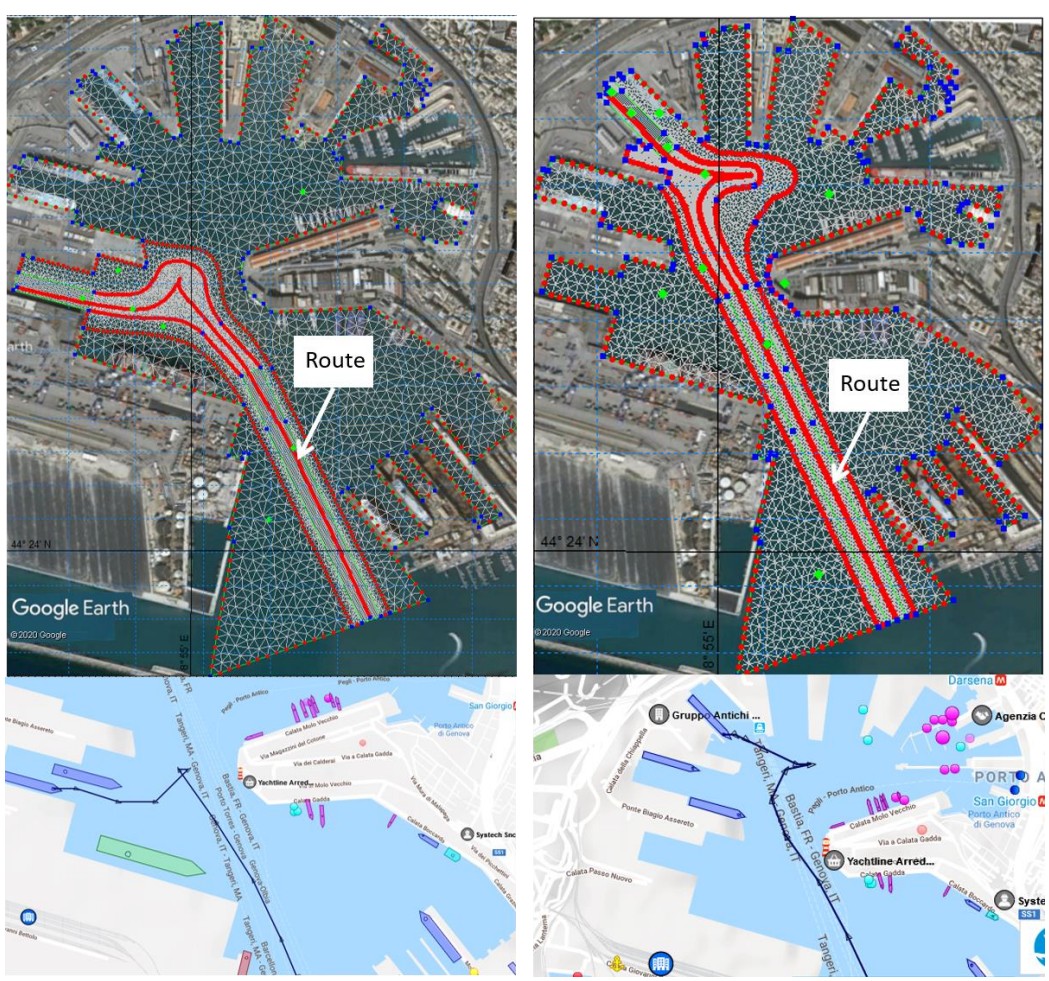

**Figure 4 - Model domain and computational grids for docking routes of T1 (left panel) and T10 (right panel)**

**docks. In the lower panels the corresponding actual routes are shown**

**Table 1 - Analysis of ship traffic in the port of Genoa for year 2017 and main characteristics of the ship representative of each dock. The ship's length, width, draught and propeller's diameter values are expressed in meters**

| Dock | Number of Berthing | % Berthing | Average Length [m] | Average Width [m] | Average Draught [m] | Average Diameter [m] |
|---|---|---|---|---|---|---|
| 1012 | 122 | 6.4% | 318.41 | 37.86 | 8.33 | 5.80 |
| 1003 | 47 | 2.5% | 276.20 | 30.07 | 7.45 | 5.20 |
| D.L. | 12 | 0.6% | 290.86 | 32.02 | 7.82 | 5.40 |
| T11 | 123 | 6.4% | 213.23 | 31.67 | 7.16 | 5.20 |
| T10 | 202 | 10.5% | 181.88 | 26.44 | 6.46 | 4.70 |
| T9 | 8 | 0.4% | 152.96 | 24.81 | 5.91 | 4.40 |
| T7 | 308 | 16.1% | 214.27 | 26.45 | 6.85 | 4.90 |
| T6 | 291 | 15.2% | 204.93 | 26.35 | 6.62 | 4.80 |
| T5 | 351 | 18.3% | 203.93 | 29.57 | 6.95 | 5.00 |





| | | | | | | |
|---|---|---|---|---|---|---|
| **T3** | 87 | 4.5% | 155.16 | 25.60 | 6.17 | 4.50 |
| **T2** | 202 | 10.5% | 185.66 | 27.85 | 6.68 | 4.80 |
| **T1** | 164 | 8.6% | 204.00 | 28.33 | 6.93 | 5.00 |
| **TOTALE** | **1917** | 100.0% | --- | --- | --- | |


In the vertical, the model is resolved over 10 sigma layers evenly distributed. The resulting layers depth vary from
approximately 1 meter in the berthing areas to approximately 2 meters in the pits and in the areas closer to the port's
entrance.
**4.1.1 - Propeller's jet velocity**
The propellers maximum jet velocity was calculated through the guidance provided in the Code of Practice of the
Federal Waterways Engineering and Research Institute (Abromeit et al., 2010) and in the PIANC Report n. 180
(MarCom WG 180, 2015), basing on the German approach. The relevant parameters for the calculations are those
shown in Figure 1. The maximum velocity $V_0$ after the jet contraction generated by the propeller is developed along the
propeller's axis. For unducted propellers it is described by Eq. (1a), for propeller ratio $J=0$ (ship not moving) or Eq.
(1b) for $J\neq0$ (moving ship).

$$V_0 = 1.60 f_n\, n_d D \sqrt{K_T} \tag{1a}$$

$$V_{0j} = \frac{\sqrt{(J^2 + 2.55 K_{Tj})}}{\sqrt{1.4\frac{P}{D}}} V_0 \tag{1b}$$

where $n_d$ [1/s] is the design rotation rate of the propeller, $f_n$ is the factor for the applicable propeller rotation rate (non
dimensional), $D$ is the propellers diameter [m], $K_t$ or $K_{tj}$ is the thrust coefficient of the propeller (non dimensional) in the
case of non-motion or motion of the ship, respectively; $P$ is the design pitch [m]. Typical values for $fn$ are 0.7 - 0.8
during manoeuvring activities, while the $P/D$ ratio can be assumed approximately equal to 0.7. $K_t$ or $K_{tj}$ can be estimated
through Eq. (2a) and (2b), according to the state of motion of the ship:
$$K_t = 0.55\frac{P}{D} \tag{2a}$$

$$K_{tj} = 0.55\frac{P}{D} - 0.46J \tag{2b}$$

The propeller ratio $J$ depends on a wake factor $w$ varying from 0.20 to 0.45 (non-dimensional) and on the velocity of the
ship according to Eq. (3):
$$J = \frac{v_{s(1-w)}}{nD} \tag{3}$$



As proposed by Hamill (Hamill, 1987) and further described by Wei-Haur Lam et al. (Lam et al. 2005), the downstream
propeller induced jet is divided into a zone of flow establishment (closer to the propeller) and a zone of established flow
(further downstream). The resulting velocity $V_0$ used in the model to calculate the corresponding discharge and
momentum sources is considered as the maximum velocity at the beginning of the zone of the established flow.
Having no direct information on the size of the ship's propellers, reference was made to specific literature on this topic.
In particular, for what concerns the propellers of Ro-Ro ferries which normally serve docks T1, T2, T3, T5, T6, T7, T9,
T10 and T11, we relied on the report n° 02 of the project "Mitigating and reversing the side-effects of environmental
legislation on Ro-Ro shipping in Northern Europe" (Kristensen, 2016) implemented by the Technical University of
Denmark (DTU) and HOK Marineconsult ApS. According to this study the relationship between the draught and the
diameter of the ferry's propeller is given by Eq. (4):
$$D_{prop} = 0.56 \; x \; H_{draught} + 1.07 \qquad (4)$$

where $D_{prop}$ is the propeller's diameter [m], and $H_{draught}$ is the maximum draft of the ship [m]. Such relation is not valid
for cruise ships having usually bigger propellers. For this type of ships, serving docks 1012, 1002 and, only partially,
D.L. and T11 we relied on direct communications from operators passenger ships design sector, and double checked the
information with formulas of Eq. (4) and Eq. (5), this latter valid for double propeller passenger ships. This qualitative
analysis brought to the diameters presented in Table 1.
$$D_{prop} = 0.85 \; x \; H_{draft} - 0.69 \qquad (5)$$
In order to represent the propeller in a realistic way the water discharge obtained combining the diameter of the
propeller and the intensity of the jet is discretized into a certain number of smaller discharges respectively associated in
the numerical model to different smaller sources of momentum. The distribution of volume and momentum sources
follows a Guassian (normal) distribution with a discretization step of 0.5 meters.
Figure 5 shows the representation of the propeller's induced jet in the hydrodynamic model. The left panel represents
the plan of Dock 1012, where a large cruise ship is departing. The solid line of the upper left panel is the location of the
vertical transect shown in the upper right image, representing the jet velocity in the plane $xz$. The dashed line in the
upper left panel represents the trajectory followed by the axis of the departing ship, and the associated jet's velocity in
the $yz$ plane is shown in the bottom panel. Albeit the non optimal horizontal resolution in terms of propellers
representation, the resulting jet appears extremely realistic both in the transverse and in the longitudinal directions.





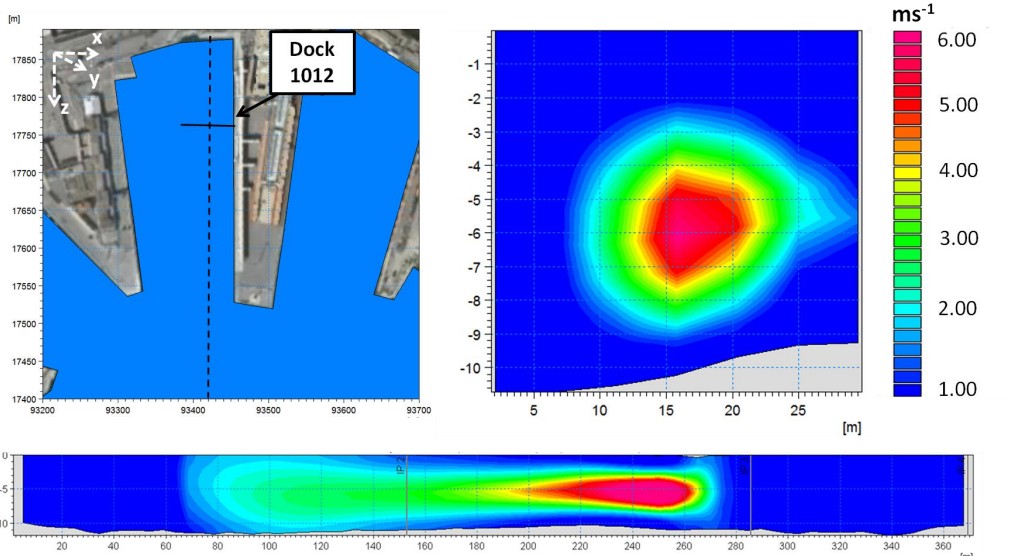

**Figure 5 – Representation of the propeller-induced jet of the most representative ship departing from Dock 1012. Left: plan view; the dashed line represents the trajectory followed by the axis of the undocking ship, the solid line represents the position of the vertical transect shown in the upper right panel, showing the jet's induced velocity in the *xz* plane (propeller's plane). Lower panel: transect of velocity along the propellers axis (*yz* plane). Velocities are in ms⁻¹**

In order to preserve the water mass budget we associated a sink to each source. Sinks are prescribed in terms of negative equivalent discharge ($m^3s^{-1}$) in the adjacent grid cell to the one hosting the source, in the direction of the ship motion (sinks precede corresponding sources).

The choice of the vertical and horizontal resolution of the hydrodynamic model was the result of a thorough sensitivity analysis to the grid's cells dimension. We assumed that the most appropriate resolution for the model is the one that allows the maximum (jet centreline) current produced by the combined discharge and momentum sources in the model to reach the input maximum velocity $V_0$. For the sensitivity analysis we considered a 4-meter diameter propeller with rotation rate of 2 rounds per second (rps) at full power. According to Eq. (1b), such a configuration results in a $V_0$ of approximately 6 ms⁻¹ at the depth of the propeller's axis once the jet is fully developed. To this purpose, we set up an experimental configuration domain, 100 meters wide and 500 meters long. The different horizontal resolutions tested were 20 m, 10 m, 5 m, 2 m and 1 m, while for the vertical we considered two configurations: 10 and 20 layers in a constant bathymetry of 20 meters. The input value of the jet current to the model was 6 ms⁻¹.



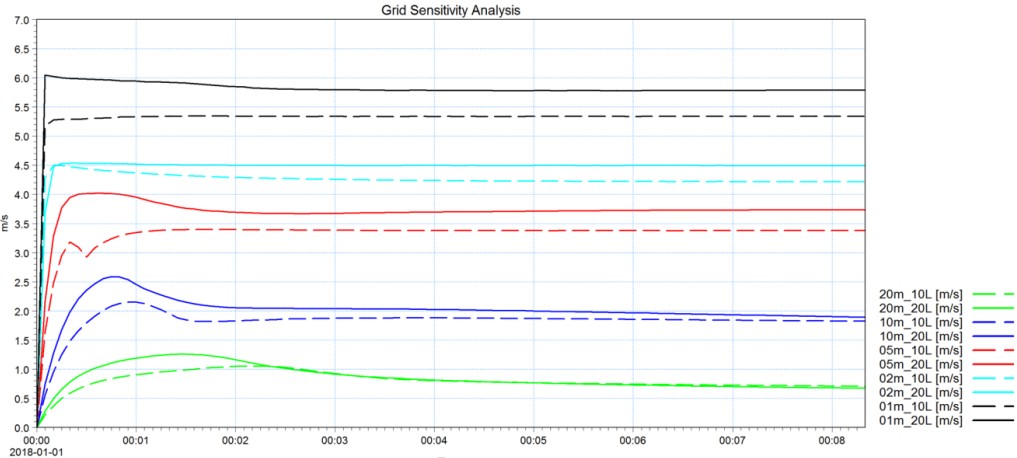


**Figure 6 – Model grid sensitivity analysis to the cells dimension. The different colors correspond to the different**


**horizontal resolution. Dashed lines indicate the configurations with 10 layers while solid lines indicate those with**


**20 layers**


Figure 6 shows the sensitivity analysis to the grid resolution. The resulting velocity at the propeller's axis is
proportional to the resolution, both in vertical and in the horizontal: the higher the resolution, the higher the resulting
velocity. The most appropriate grid would be the one with 1 meter resolution and 20 vertical layers, which is the only
configuration of the model which allows the jet to reach the maximum speed imposed as input. However, this
configuration would require approximately one year of computational time to run the 24 simulations implemented for
this study in the same computational configurations, which is obviously unrealistic. We thus looked for a compromise
between acceptable computational demand and realistic resulting velocity. The final configuration was the one with 5
meters as horizontal resolution and 10 vertical levels. Since such resolutions would not allow the complete development
of the current speed we introduced a correction to the input velocity of each simulated vessel by increasing it of the
necessary amount to reach the empirically calculated $V_0$. This implied considerable additional time for manual
calibration.
**4.1.2 – Forcing and boundary conditions**
Due to the nature of the processes of interest the only forcing accounted for is the propeller of the vessels. In fact, the jet
induced by its motion is of the order of magnitude of several meters per second in the surrounding of the blades, and it
has a length of influence of at least 40-50 times the propeller's diameter behind the ship (Verhei, 1983). Natural forcing
such as wind, density gradients or tides are one to two orders of magnitude smaller in this area, thus they can be
neglected without introducing errors potentially impacting on sediment resuspension from the bottom. On the contrary,
Bernoulli wake might be responsible for currents of comparable intensity (Rapaglia et al., 2016), albeit smaller, and it



would be worth to be considered as a forcing of the system. In this study, though, we neglected it due to technical
complications and time obligations. It will be interesting to include such process in further developments and to analyse
the impact on the overall dynamics of ship induced sediment transport. However, the satisfying final results of the
present work suggest that the governing processes for these dynamics are associated to propellers induced currents more
than to the motion of the ship itself, likely due to the limited vessels speed in this inner part of the harbour and to the
relatively large volume of water available for each passing vessel.
The boundaries of the hydrodynamic domain are the docks all around the basin and the port entrance, which is the only
open boundary. Here we imposed a Flather condition (Flather, 1976) assuming constant zero velocities and levels. Such
a choice allowed to minimize the boundary effects, albeit some interference between the flux and the boundary line is
present (not shown). However, due to the distance between the open boundary line and the berthing areas such effects
do not influence the results of the study. A zero normal velocity was imposed along the closed boundaries.

**4.2 – The sediment transport model**
The hydrodynamic model was coupled with a sediment transport model – MIKE 3 MT FM - valid for fine-grained and
cohesive sediment (diameter smaller than 63 μm, Lisi et al., 2017). This type of sediment is mostly present in the port
of Genoa and particularly relevant for the erosion, transport and further deposition, since its small dimension and weight
favour relevant resuspension and advection around the basin.
The governing equations of the mud transport model are based on the advection and dispersion (AD) of the
concentration of the sediment along the water column and they are detailed in APPENDIX A2. The AD equation is
solved using an explicit, third order finite difference scheme called ULTIMATE (Leonard, 1991).
The model accounts for two compartments: a water and a seabed environment. The seabed is represented through a
multi-bed layer and multi-fraction approach in which the different layers can exchange mass and only the top level is
active, thus available for erosion. The different layers are defined through the fractions of sediment they're composed
of, the degree of consolidation of the sediment within each layer, and the thickness of the single layer. The different
sediment fractions are described through their associated physical characteristics, and they are eroded and deposited
proportionally to their concentration both in the bed texture and along the water column. Within the water environment,
the model includes flocculation processes when exceeding a certain threshold of concentration (here assumed equal to
0.01 gl-1) and hindered settling according to Wintwerp (Winterwerp and Van Kesteren, 2004) definition with a
threshold of 10 gl-1. The deposition of the sediment is based on a Teeter (Teeter, 1996) profile and the threshold for
deposition used was 0.07 Nm$^{-2}$. The sediment grain diameter is defined through the associated settling velocity, based
on Stokes law. In the interface between the water and the bottom the sediment may be eroded following the approach by





Partheniades (Partheniades, 1965) for consolidated sediment or that by Parchure and Metha (Parchure and Metha, 1985)
for soft or unconsolidated sediment. In both cases the sediment is eroded and injected into the water column when the
shear stress resulting from the current, the wave action or a combination of both exceeds a certain critical value. In the
present case waves were not considered since we are inside the port.
The specific equations and parameterizations referred to in the sediment model are summarized in APPENDIX A2.

**4.2.1 - Sediment characteristics**
Three different sediment surveys were carried out between June 2009 and July 2010. Table 2 presents the results of the
surveys in terms of percentage and class of sediment per survey (last and central column, respectively). Given the
nature of the study we are interested in mud and fine sand, thus the part of the texture coarser than 2 mm was not taken
into consideration.
**Table 2 - Sediment size data inside the port (see station identified with the red dot of Figure 2). Three different**
**surveys were carried out between June 2009 and July 2010**

| Date of survey | Sediment Size | % |
|---|---|---|
| 2009-06-15 16:00:00 | Ø < 63 μm | 82.4 |
| 2009-06-15 16:00:00 | 63μm < Ø < 2mm | 16.2 |
| 2009-06-15 16:00:00 | Ø > 2 mm | 1.4 |
| 2009-07-15 16:00:00 | Ø < 63 μm | 89.2 |
| 2009-07-15 16:00:00 | 63μm < Ø < 2mm | 9.1 |
| 2009-07-15 16:00:00 | Ø > 2 mm | 1.7 |
| 2010-07-28 09:00:00 | Ø < 63 μm | 78.2 |
| 2010-07-28 09:00:00 | 63μm < Ø < 2mm | 17.7 |


We assumed that the fraction of the samples with Ø < 63 μm was composed by two grain sizes with diameters of 30 *μm*
and 50 *μm* respectively, while for the observed component with diameter in the range of 63*μm* to 2 *mm* we assumed the
diameter of 100 *μm* would be representative for the present study.
The three fractions chosen were distributed into three active bed layers. The percentage of the fine fractions amongst the
texture of sediment was assumed to decrease proportionally to the depth of the layers. Thus, the first layer contained
80% of fines (specifically 50% of grains with Ø=30 *μm* and 30% with Ø=50 *μm*) and 20% of coarse (Ø=100*μm*), while
the third layer contained 50% of coarse (Ø=100*μm*) and 50% of fines (specifically 20% of grains with Ø=30 *μm* and
30% with Ø=50 *μm*). In the mid layer an even distribution was assumed among the three fractions. The thickness of the
three layers is 0.5 *mm*, 1 *mm* and 50 *mm* at the beginning of each scenario. The first layer is composed by very soft mud
since it is the result of the newly deposited and finer mud. The other two layers are more consolidated and thicker, since
they are harder to be eroded and they are shielded by the upper layers. The adopted description of the bottom with



different layers and fractions of sediment allowed to represent the port bed in a complex and comprehensive way,
including the different degree of consolidation of the layers and the resulting different response to shear stress
solicitations.
A summary of the most relevant characteristics of the layers and sediment fractions implemented in the sediment
transport model is presented in Table 3.
Finally, potential sediment input might come from six minor streams inflowing in the port area. They have very modest
basins - approximately 1 km² on the average – and they have been ceiling-covered for long time, acting now more as
sewage collectors than as natural streams. An estimate of their contribution to the sedimentary dynamics of the port of
Genoa has been conducted and the annual sediment supply to the port basin from each stream has been evaluated
referring to the method proposed by Ciccacci et al. (Ciccacci et al., 1989). The estimated contribution of sediment
resulted in only a few hundreds of cubic meters per year in the worst cases, which corresponds to a few millimetres of
annual accumulated sediments in the surrounding of the river inlet to the wet basins. Such amount of solid matter has
not been considered in the model since the erosional and depositional processes induced by the propellers' activity are
higher by one or two orders of magnitude.
**Table 3 – Summary of sediment characteristics as implemented in the mud transport model**

| Parameter | Layer 1 | Layer 2 | Layer 3 |
|---|---|---|---|
| Layer thickness (mm) | 0.5 | 1 | 50 |
| Type of Mud | soft | hard | hard |
| Dry density of bed layer (kgm$^{-3}$) | 180 | 300 | 450 |
| **Parameter** | **Fraction 1** | **Fraction 2** | **Fraction 3** |
| $\Phi$ ($\mu$m) | 30 | 50 | 100 |
| % of fraction in layer 1, 2, 3 | 50, 33, 20 | 30, 33, 30 | 20, 33, 50 |
| $W_s$ (mms$^{-1}$) | 0.7 | 2.2 | 8.8 |
| $\tau_{ce}$ (Pa) | 0.15 | 0.25 | 0.5 |
| $\tau_{cd}$ (Pa) | 0.07 | 0.07 | 0.07 |
| $C_{floc}$ (gl$^{-1}$) | 0.01 | 0.01 | 0.01 |
| $C_{hind}$ (gl$^{-1}$) | 10 | 10 | 10 |
| $\rho_s$ (kgm$^{-3}$) | 2650 | 2650 | 2650 |



**5 - Results**
The most representative results of the hydrodynamic and sediment transport model are presented in this section. Due to
the large number of simulations carried out, only those regarding two docks are shown. However, the results not shown
corresponding to the other simulations are similar in terms of hydro and sediment dynamics. The results discussed are
those of the simulations of docks 1012 and T7. Dock 1012 is particularly important since it hosts the biggest passenger
vessels operating in the port, while dock T7 is particularly relevant due to the high frequency of passages.





Figure 7 shows the propeller's generated current in the bottom layer and at the depth of the propeller's axis (upper right
and left panels, respectively) and the resulting suspended sediment concentration in the same layers (corresponding
lower panels) during the departure of a cruise vessel from dock 1012. The characteristics of the vessel representative of
the traffic of the dock are those of Table 1. When departing, the engine is operated close to full power, which we
assumed to result in a rotation rate of 2 rounds per second (rps) for the propeller. This induces a maximum velocity at
the depth of the propeller axis close to 9 ms$^{-1}$ which is damped to approximately 2 ms$^{-1}$ on the bottom of the berthing
basin along the vessel's route. Such intense jet is deflected to the left due to the head wall of the berthing basin which
constrains the flow and induces a cyclonic eddy, well developed along the whole water column. The cone-like envelop
of the jet in the vertical plane as sketched in the theoretical scheme of Figure 1 is appreciable from the upper panels of
Figure 7, which refer to the same instant: the influence of the propeller on the bottom occurs several tens of meters
behind the propeller's position, and the velocity at the bottom is strongly reduced. The induced eddy in the wet basin
acts as a trap for the eroded sediment, which enters the cyclonic gyre (or anti-cyclonic in the case of departure from the
opposite dock) and tends to deposit in the middle of the basin, where the fluxes progressively decrease. The position of
the eye of the cyclone evolves parallel to the docks' longitudinal walls and induces the sediment trapped inside the gyre
to sink along the longitudinal axis of the wet basin. Such dynamic occurs similarly for all the horseshoe-shaped wet
basins, inducing accumulation along the central portions. The re-suspended sediment may reach very high
concentrations in the bottom layers, up to several hundreds of mgl$^{-1}$, depending on the different specific characteristics
of the sediment texture (mainly grain size, level of compactation, availability to erosion) and of the vessel (mainly
dimension of the propellers, rotation rate, draught).
Different hydro and sediment dynamics occur during the inbound phase of vessels manoeuvring inside the port. The
majority of the manoeuvring operations (i.e. when vessels rotate within a turning basin and proceed backwards to the
docks) occur in the turning basins delimited by the dashed circles *a* and *b* shown in Figure 2. When starting the
manoeuvre, engines operate to high power in order the allow the rotation of the ship. Within these operations the
vessel's longitudinal axis rapidly changes direction (order of tens of seconds up to a few minutes) spanning wide angles
according to the specific manoeuvre to be undertaken. The propellers induced jet follows the same rotation along the
horizontal plane resulting in a fan-like distributed set of directions for the associated currents. Such operations are
represented by the model in a realistic way as shown in Figure 8, which refers to the berthing of vessel representative of
dock T7. The currents shown in the figure are those associated to the propeller's axis during four different moments of
the turning manoeuvre. Each panel refers to a time interval of approximately 100 seconds from the previous one. The
successive instants are presented in the order up-left, up-right, down-left and down-right, respectively. In the lower-
right panel the propeller has already changed direction of rotation and the vessel is now proceeding backwards. The



induced current jet is thus heading towards the centre of the port, pushing the sediment towards this area. What
simultaneously happens at the seabed is shown in Figure 9. Albeit the jet induced currents are very much weaker at the
seabed than those at the depth of the propeller's axis, they are still relevant and may reach intensities up to 1 ms$^{-1}$,
depending on the local bathymetry.

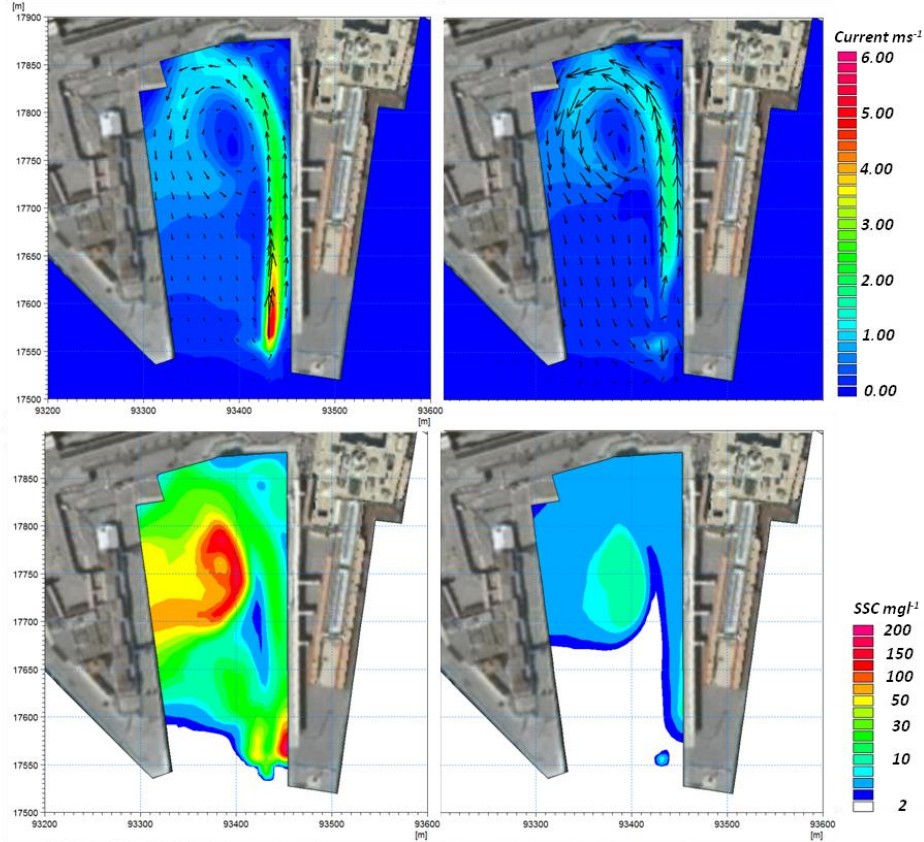


**Figure 7 – Results of the numerical models. Upper panels: current intensity and direction in the bottom layer**
**(right) and in the layer corresponding to the axis propeller. Lower panels: resulting suspended sediment**
**concentration (SSC, mgl$^{-1}$) in the same layers as the upper panels. The images refer to the undocking of the**
**cruise vessel representative of dock 1012.**
The current distribution at the seabed is much more chaotic than at the propeller's axis depth. It is to be noted that this
area of the port corresponds to the natural pit (which reaches 22 meters below the surface in the deeper part,
approximately) where the material dredged from accumulation areas is normally dumped during the sea bottom
maintenance activities. The dashed line shown in the lower-right panels of Figure 8 and Figure 9 refers to the transect



presented in Figure 10, in the same instant (i.e. when the vessel has ended the manoeuvre in the circle *b* and is
approaching dock T7 backwards).

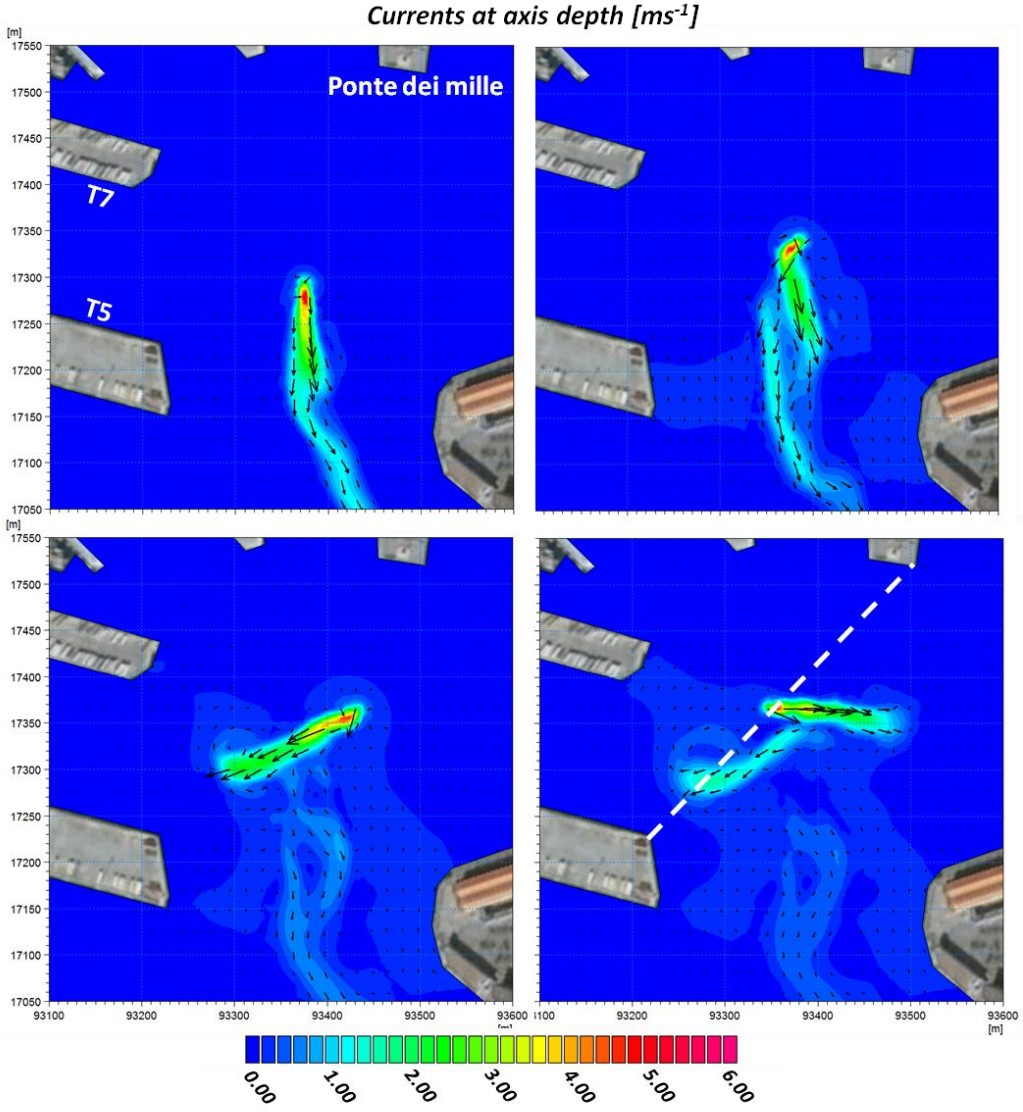


**Figure 8 – Results of the hydrodynamic model at the depth of the propeller's axis. Each panel refers to a time**

**interval of approximately 100 seconds from the previous one. The temporal order of the panels is up-left, up-**
**right, down-left and down-right**



A combined analysis of Figure 8, Figure 9 and Figure 10 helps understand the dynamics occurring in the turning basin *b*
during manoeuvres to approach docks T5, T6 and T7. This is particularly important in order to understand the overall
sediment dynamics of the entire port since these three docks operate approximately half of the entire passenger traffic.
The propeller's induced velocities at the bottom of the natural pit during turning manoeuvres is variable and may
exceed 1 ms$^{-1}$, which is a relevant current intensity able to entrain and move a large amount of sediment. The resulting
re-suspended sediment concentration may reach important values, exceeding 50-60 mgl$^{-1}$, as shown in the lower panel
of Figure 10. Once re-suspended from the pit, the sediment is advected around by the jet induced complex field of
currents of Figure 8 and Figure 9. This area is normally refilled with freshly dredged material resulting from the seabed
maintenance activities, thus the propeller's induced currents on the bottom have an enhanced effect of erosion on the
unconsolidated material and are able to rapidly nullify the benefit of the dredging operations. In this regard, the results
of the simulations suggest to avoid to use the natural pit as a dumping area for the resulting material of such activities
and prove that integrated modelling can be a fundamental tool for the comprehension of the processes and mechanisms
related to sediment transport and for an optimized planning of maintenance activities.







**Figure 9 – Same as Figure 8 but for the bottom layer**



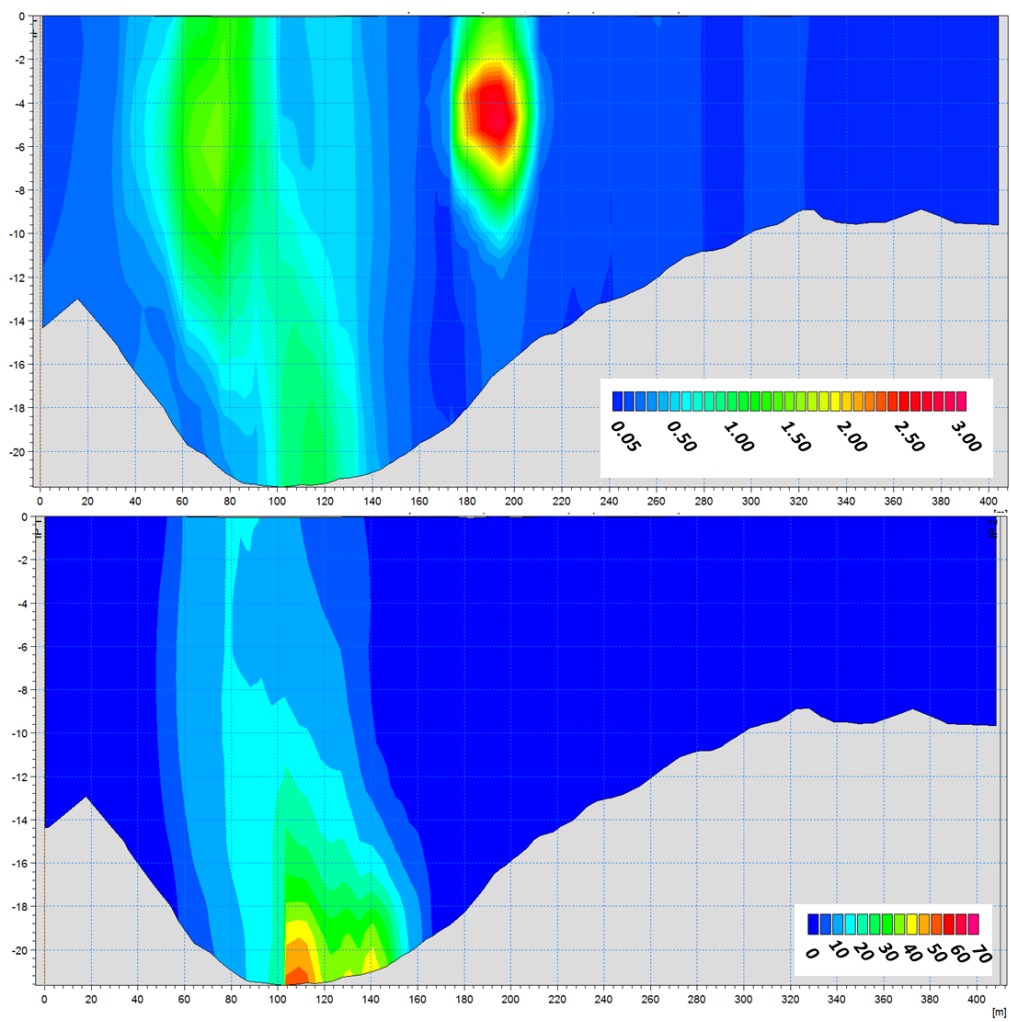

**Figure 10 – Velocity intensity in ms⁻¹ (upper panel) and sediment concentration in mgl⁻¹(lower panel) along the**

**transect from the head of *Ponte Assereto* to the head of *Ponte dei Mille***

The impact on the bed thickness due to the naval traffic is depicted in Figure 11, which presents the erosion and

deposition maps resulting from the simulations of one departure (left) and one arrival (right) of the representative

passenger vessels of docks 1012 (up) and T7 (down). The blue colors represent areas of erosion, while the red colors

represent those of accumulation of the sediment after an interval of time sufficiently long for the re-suspended sediment

to completely settle down. It is evident from the left panels of the figure that during the vessel's departure a

considerable amount of material tends to be eroded from the basement of the docks and settles in the center of the

mooring basins. This mechanism is clearly related to the vessel's departure (left panels) rather than to the vessel's

arrival (right panels). The erosion underneath the vessel's keel along the ship's trajectory is well evident both during

departure and arrival., in agreement with experimental literature findings (Castells et al. 2018). The order of magnitude
of erosion and deposition of one single vessel's passage is of a few millimeters in the areas most influenced by the
vessel's activity.

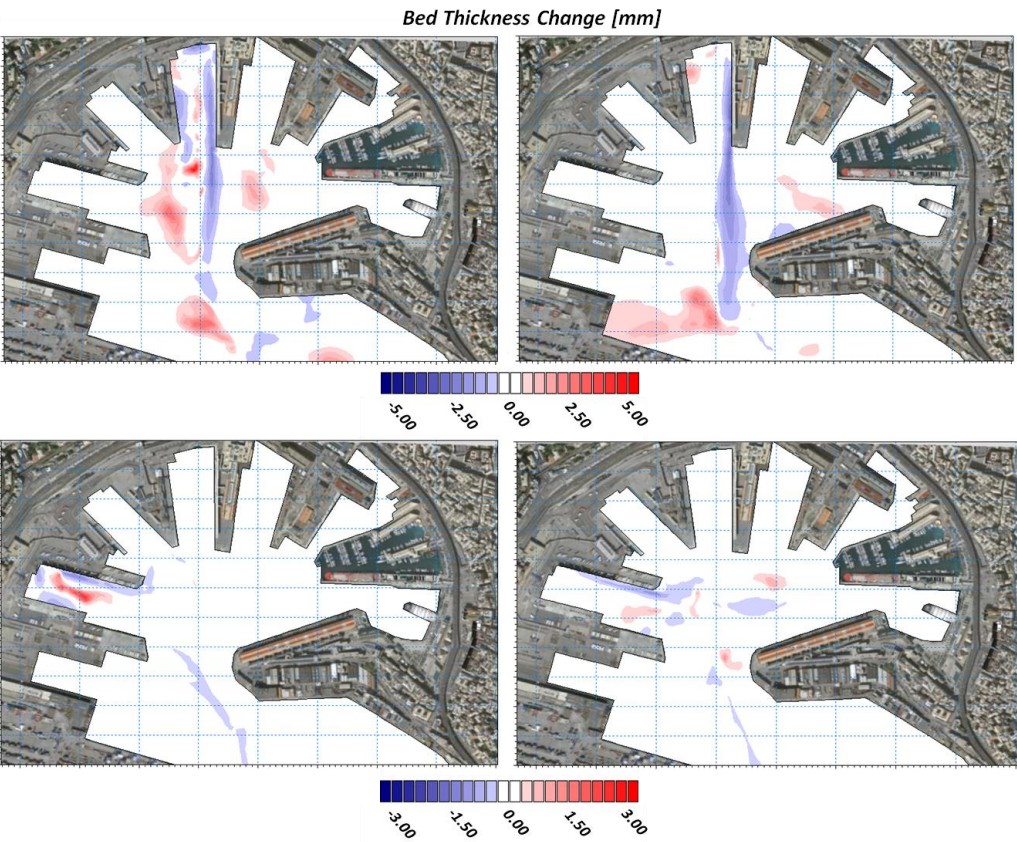


**Figure 11 – Erosion and deposition maps resulting from one departure (left) and one arrival (right) of the**
**representative passenger vessels of docks 1012 (up) and T7 (down)**
Such impact might become a real threat for the continuity of the operability of large and busy ports like the Port of
Genoa in the medium and long timescales. The few millimeters of accumulation and erosion might become several tens
of centimeters after a few thousands of annual passages. Relying on the traffic analysis of Table 1 we projected each
single naval passage to a one-year period and superimposed the effects of erosion and deposition of the vessels
representative of all the passenger docks. We were thus able to reconstruct the annual port seabed evolution for year
2017. The effects of the single passages were weighted by the occurrences of the year 2017, thus obtaining 24 maps
(one for each docking and one for each undocking), and the results of the 24 maps were integrated to obtain the final





491 map. To take into account the fact that the trajectories to reach a dock (or to depart from it) slightly vary from passage

492 to passage, a Bartlett spatial filter was applied to the integrated results using the values 4, 2 and 1 as weights. Figure 12

493 presents the results of this analysis. In the left panel the results from the modeling system in terms of annual erosion

494 (blue) and accumulation (red) are shown, while in the right panel the observed seabed evolution is shown. The observed

495 map was reconstructed through the results of two different bathymetric surveys carried out in the periods May-June

496 2017 and March-June 2018. The difference of the bathymetries of the two surveys resulted in the evolution of the

497 seabed during the approximate period of one year, except for dredging operations. We used numbers in the maps to

498 indicate areas where the most relevant dynamics outlined by the study take place.

499 It is to be noted that the area between the head of *Ponte dei Mille* and the head *of Molo Vecchio* identified as 1 was

500 dredged during the period October-December 2017 and approximately 15.000 $m^3$ of solid material were removed and

501 dumped into the natural pit of the port, here indicated with number 5. Consequently, what appears at first sight from

502 observations as an area of erosion due to the vessel traffic - area 1 in the right panel of Figure 12 - is actually an area of

503 accumulation, as it is also confirmed by the fact that dredging operations were conducted. Similarly, the accumulation

504 observed in area 5 (right panel of Figure 12) is not the result of the induced action of the propellers, but it is the result of

505 the accumulation of the sediment dumped after maintenance dredging operations. The model results are in total

506 agreement with these dynamics. As discussed above, the material re-suspended during vessels' maneuvers is likely

507 pushed towards area 1 during the phase of backward advancing of the vessels when approaching the docks. On the

508 contrary, area 5 is partially an area of erosion, as evidenced by the model. The freshly deposited material during

509 dredging operations is thus soon re-suspended.



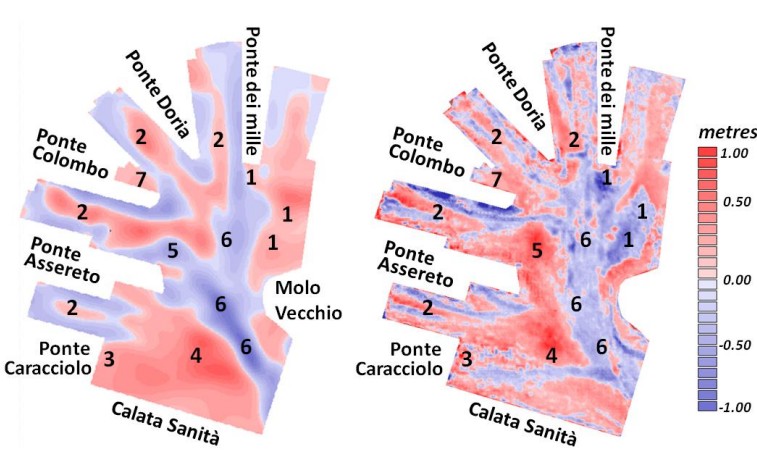

**Figure 12 – Annual erosion and deposition map reconstructed on the basis of the hydrodynamic and sediment transport simulations for the year 2017**

Area 1 accounts for approximately 30-40 cm of accumulated material per year, with local maxima of up to 50 cm. Similar values were estimated against years of managing experience by personnel of Stazioni Marittime S.p.A (personal communication).

The central portions of the wet basins marked with number 2 in Figure 12 are areas of deposition, mainly due to the phase of departure of the ships. Again, the model is able to well reproduce both the accumulation along the central parts of the basins, where it may reach 20 cm per year or even more, and the erosion along the walls of the docks. Here the propellers erosive action might result in issues for the stability of the docks, especially along those of dock 1012, where the biggest cruise vessels operate.

The erosion underneath the vessels' typical routes (i.e. from the entrance to approximately the center of the port) is also well represented by the model, and it is identified in the figure with the number 6. Good agreement between the model and the observations is also evident in the deposition area identified with the number 7, where a local gyre forms and entraps the suspended sediment. Finally, also areas 3 and 4 are subject to deposition, and qualitative agreement between the model and the bathymetric differential survey is evident from Figure 12. The erosive print observed in the survey under these areas is most likely due to activities related to cargo vessels when approaching and departing from dock *Calata Sanità*. This latter was not object of the study, which was intended only for passenger docks whereas *Calata Sanità* operates only container ships, thus the model does include the naval traffic here.

In general., the comparison between the observed and the modeled annual evolution of the port seabed shows a very good agreement, it proves the reliability and robustness of the hydrodynamic and sediment transport model and it



finally shows the potential importance of an integrated modeling approach to optimize the management of the port
activities.

**5 – Summary and Conclusions**
The impact of naval traffic on the seabed of the passengers Port of Genoa was investigated by means of numerical
modeling. The combination of a very high resolution, non-hydrostatic, circulation model (MIKE 3 HD FM) with a
sediment transport model (MIKE 3 MT FM), based on unstructured grids on the horizontal and on sigma levels on the
vertical allowed to reconstruct the annual evolution of the port seabed. The final results of the modeling in terms of
maps of erosion and deposition inside the basin were qualitatively supported by observational evidence. The approach
followed was to simulate only one arrival and one departure from each dock of the port and to analyze the impact of a
single naval passage on the seabed in terms of sediment concentration, motion and distribution.
Following the traffic analysis in the port for a typical year (year 2017) the detailed situation of the number of arrivals
and departures for each dock was available as a starting point for the study. Through the superimposition of the single
effects of the traffic weighted for the annual number of passages of the most representative vessel operating on each
dock the annual map of erosion/deposition was reconstructed and validated on a semi-quantitative basis versus
differential bathymetric surveys available for the same period.
In general., the simulations showed that the velocity intensities on the bottom induced by propeller's generated jets may
reach almost 2 ms$^{-1}$, mainly depending on the dimension of the propellers, on the rotation rate and on the distance
between the propeller and the bottom. Such velocities may reach up to 8-9 ms$^{-1}$ at the propeller's axis depth, and
penetrate horizontally through the water for long distances, up to at least 40-50 times the propeller's diameter. The bed
shear stresses induced by these velocities, as well as the propeller jet induced entrainment mobilize and re-suspend high
amounts of the fine and less compacted sediments present inside the port. Fine fractions with smaller fall velocities tend
to remain in suspension for longer periods of time, resulting in creation of sediment plumes. Hong et al. (2016) have
shown in their laboratory test results the dependency of the concentration profiles behind propeller jets to sediment
grain size distribution, amongst other parameters.
The final findings showed how relevant the deposition rates might be in a densely operated port, reaching values of
several tens of centimeters per year in some local areas.
The type of approach we adopted was particularly useful not just because it allowed to minimize the computational
time, but also because it allowed to decompose the overall complex picture of sediment transport of the entire port into
several simpler pictures. Consequently, the analysis of the single hydro and sediment dynamics occurring for each dock
and vessel was possible as well as the identification of the specific routes responsible of the particular problems of





erosion and accumulation historically reported by the managing authorities of the port operations and traffic. The range
of current intensities induced by the propellers action was identified along the water column, and it can be further used
as a solid and scientific-based benchmark value for potential defensive actions for the seabed and port structures that
might be undertaken in the future in order to preserve the port's full operability.
The most relevant mechanisms regarding the port hydro and sediment dynamics occurring during vessels passages were
identified and the following analysis allowed to understand how and why specific areas are subject to erosion and other
areas are subject to deposition, and to what extent these mechanisms occur. In particular, the mechanism of erosion
ongoing along the docks walls and that of deposition along the central portions of the mooring basins were identified
and explained, as well as the deposition process constantly ongoing in the area confined between the head of *Ponte dei*
*Mille* and the head of *Molo Vecchio*. This last process was particularly important to reproduce and understand for the
port managers since it occurs at a very important rate, up to 40-50 cm per year in some local areas. Finally, the natural
hole located off the heads of *Ponte Colombo* and *Ponte Assereto* was identified through the model as an area of erosion,
albeit its relevant depth. This is mainly due to the turning maneuvers carried out by vessels in this area which partially
corresponds to one of the turning basins of the port and which involves approximately the 50% of the entire traffic of
the port (docks T5, T6 and T7). Since such location has been historically used as a dumping site for the resulting
material of seabed maintenance dredging, the study showed how unfit this area is for such purpose, since the freshly
deposited sediment is soon re-suspended by the intense currents induced by the vessels turning operations.
The importance of this study was not only to prove how integrated high resolution modeling might be able to reproduce
the most relevant and complex mechanisms of hydrodynamics and sediment transport occurring inside ports – which
was however done successfully – but it was also to suggest, once its reliability was proven, that it can be used as a
fundamental tool for an optimized port management. In fact, it could be used to regulate the naval traffic in ports in
order identify the most suitable schedule and routing in terms of sediment concentrations, bottom velocities, erosion and
accumulation. Or again it could be used to identify the biggest vessels potentially operating in the docks for the
planning of the future commercial traffic, or to study the impact of the increasing traffic of ports on the seabed and on
the ports structures, or finally for an awareness planning of the recurring dredging operations related to the sediment
accumulation problems that the majority of densely operating ports must regularly face, most of the times without being
correctly prepared.
Daily fully-operational implementations of similar integrated systems are also possible to set up, since the daily
schedule of the port is known. This would allow to continuously monitor the evolution of the seabed and to be
constantly and fully aware of the potential criticalities to face.





An important process that should be included in the future developments of the present study is the effect on the
sediment resuspension, advection and dispersion due to Bernoulli wake and its combination with the propeller's
induced jets. This mechanism was not included in the present version of the system. The current intensities caused by
vessels' generated waves during and after their passages are surely smaller than those induced by propellers along their
axis, but they tend to penetrate along the water column and reach the bottom carrying a significant amount of energy,
and possibly re-suspending important amount of solid material (Rapaglia et al.2011), probably enhancing the vertical
mixing and maybe inducing the sediment to be suspended for longer periods and at higher depths.


**APPENDIX A1 – Hydrodynamic model governing equations**
MIKE 3 Flow Model FM is based on the Navier-Stokes equations for an incompressible fluid under the assumptions of
Boussinesq. The governing equations of the model are the equations of momentum (A1.1) and mass continuity (A1.2),
the equations of heat and salinity transport (A1.3 and A1.4, respectively) and the equation of state (A1.5) based on the
UNESCO formula of 1981 (UNESCO, 1981a). Considering a Cartesian coordinate system *(x,y,z)* we have:
$$\frac{\partial u}{\partial x} + \frac{\partial v}{\partial y} + \frac{\partial w}{\partial z} = 0 \tag{A1.1}$$


$$\frac{\partial u}{\partial t} + \frac{\partial u^2}{\partial x} + \frac{\partial vu}{\partial y} + \frac{\partial wu}{\partial z} = fv - \frac{1}{\rho_0}\frac{\partial q}{\partial x} - g\frac{\partial \eta}{\partial x} - \frac{1}{\rho_0}\frac{\partial p_a}{\partial x} - \frac{g}{\rho_0}\int_z^\eta \frac{\partial \rho}{\partial x}dz + F_u + \frac{\partial}{\partial z}\left(v_t^v\frac{\partial u}{\partial z}\right) \tag{A1.2.1}$$


$$\frac{\partial v}{\partial t} + \frac{\partial v^2}{\partial y} + \frac{\partial uv}{\partial x} + \frac{\partial wv}{\partial z} = fu - \frac{1}{\rho_0}\frac{\partial q}{\partial y} - g\frac{\partial \eta}{\partial y} - \frac{1}{\rho_0}\frac{\partial p_a}{\partial y} - \frac{g}{\rho_0}\int_z^\eta \frac{\partial \rho}{\partial y}dz + F_v + \frac{\partial}{\partial z}\left(v_t^v\frac{\partial v}{\partial z}\right) \tag{A1.2.2}$$


$$\frac{\partial w}{\partial t} + \frac{\partial w^2}{\partial z} + \frac{\partial uw}{\partial x} + \frac{\partial wv}{\partial y} = -\frac{1}{\rho_0}\frac{\partial q}{\partial z} + F_w + \frac{\partial}{\partial z}\left(v_t^v\frac{\partial w}{\partial z}\right) \tag{A1.2.3}$$


$$\frac{\partial T}{\partial t} + \frac{\partial uT}{\partial x} + \frac{\partial vT}{\partial y} + \frac{\partial wT}{\partial z} = F_T + \frac{\partial}{\partial z}\left(D_{ts}^v\frac{\partial T}{\partial z}\right) + \hat{H} \tag{A1.3}$$


$$\frac{\partial S}{\partial t} + \frac{\partial uS}{\partial x} + \frac{\partial vS}{\partial y} + \frac{\partial wS}{\partial z} = F_s + \frac{\partial}{\partial z}\left(D_{ts}^v\frac{\partial S}{\partial z}\right) \tag{A1.4}$$


$$\rho = \rho(S, T) \tag{A1.5}$$

Since we used the barotropic density mode the only hydrodynamic equations used for the present work
are A1.1 and A1.2. The symbols used in the governing equations of the model are presented in Table 4




**Table 4 – Symbols used in the governing equations A1**

| $x, y, z$ | Cartesian coordinate system |
|---|---|
| $u, v, w$ | components of the field of velocity [ms$^{-1}$] |
| $g$ | gravity acceleration [ms$^{-2}$] |
| $\rho$ | water density [kgm$^{-3}$] |
| $\rho_0$ | reference value for water density [kgm$^{-3}$] |
| $q$ | non-hydrostatic pressure [Pa] |
| $p_a$ | atmospheric pressure at the sea surface [Pa] |
| $f$ | Coriolis parameter (non dimensional) |
| $\nu_t^v$ | vertical eddy viscosity [m$^2$s$^{-1}$] |
| $F_u, F_v, F_w$ | horizontal diffusivity |
| $T$ | temperature [°C] |
| $S$ | Salinity [PSU] |
| $F_T, F_S$ | Horizontal diffusion terms for $T$ and $S$ |
| $D_{ts}^v$ | vertical eddy diffusivity [m$^2$s$^{-1}$] |
| $\overset{\wedge}{H}$ | Source term due to heat exchange with the atmosphere |


**APPENDIX A2 – Mud transport model governing equations and parameterizations**
The sediment transport module is based on the advection dispersion equation for a passive tracer in an incompressible
fluid. The tracer is the concentration $C$ of sediment along the water column. The field velocity used for advection is the
one calculated through the hydrodynamic set of equations of Appendix A1. The symbols used in the set of equations A2
are summarized in Table 5
$$\frac{\partial C}{\partial t} + \frac{\partial}{\partial x}(uC) + + \frac{\partial}{\partial y}(vC) + \frac{\partial}{\partial z}[(w + w_s)C] = \frac{\partial}{\partial z}\left(D_C^v \frac{\partial C}{\partial z}\right) + F_C \qquad \text{(A2.1)}$$
The vertical bottom boundary condition for sediment flux is expressed as:
$$D_C^v \frac{\partial C}{\partial z}\Big|_{z=-H} - w_s C = S \qquad \text{(A2.2)}$$
and the sediment flux $S$ at the bottom is calculated through the approach of Krone (Krone, 1962) for deposition (Eq.
A2.3), through that of Partheniades (Partheniades, 1965) for erosion of consolidated sediment (Eq. A2.4) and through
that of Parchure and Metha (Parchure and Metha, 1985) for erosion of soft or unconsolidated sediment (Eq. A2.5).
$$S_d = w_s c_b p_d \qquad \text{(A2.3)}$$
where
$$p_d = 1 - \frac{\tau_b}{\tau_{cd}} \qquad \text{valid for } \tau_b < \tau_{cd} \qquad \text{(A2.3.1)}$$
$$S_{ec} = E\left(\frac{\tau_b}{\tau_{ce}} - 1\right)^n \qquad \text{valid for } \tau_b \geq \tau_{ce} \text{ and hard bed} \qquad \text{(A2.4)}$$


$$S_{es} = E \, exp\left[\alpha(\tau_b - \tau_{ce})^{1/2}\right] \qquad \text{valid for } \tau_b \geq \tau_{ce} \text{ and soft bed} \qquad \text{(A2.5)}$$
The settling velocity for sediment is calculated through the Stokes law (A2.6).
$$w_s = \frac{gd^2}{18}\left(\frac{\rho_s}{\rho_w} - 1\right) \qquad\qquad \text{(A2.5)}$$
**Table 5 – symbols used in the equations and parameterizations A2 of the sediment transport model**

| | |
|---|---|
| $x,y,z$ | cartesian coordinate system (same as Table 4) |
| $u,v,w$ | components of the field of velocity (same as Table 4) [ms$^{-1}$] |
| $C$ | sediment concentration [gmc$^{-1}$] |
| $C_b$ | sediment concentration in the bottom layer [gmc$^{-1}$] |
| $w_s$ | settling velocity [ms$^{-1}$] |
| $D_C^v$ | vertical eddy diffusivity for $C$ (same as for T and S) [m$^2$s$^{-1}$] |
| $F_C$ | horizontal diffusion terms for $C$ |
| $H$ | water depth [m] |
| $S_e$ | bottom sediment flux for erosion [kgm$^2$s$^{-1}$] |
| $S_d$ | bottom sediment flux for deposition [kgm$^2$s$^{-1}$] |
| $S_{e,s}$ | bottom sediment flux for erosion of soft bed [kgm$^2$s$^{-1}$] |
| $S_{e,c}$ | bottom sediment flux for erosion of consolidated bed [kgm$^2$s$^{-1}$] |
| $p_d$ | probability of deposition for the sediment [non dimensional] |
| $\tau_b$ | bottom shear stress [Nm$^{-2}$] |
| $\tau_{bd}$ | critical stress for deposition [Nm$^{-2}$] |
| $\tau_{ce}$ | critical stress for erosion [Nm$^{-2}$] |
| $E$ | bottom erodibility [Nm$^{-2}$] |
| $\alpha$ | empirical coefficient [$m/\sqrt{N}$] |
| $n$ | Power of erosion (empirical non-dimensional) |
| $d$ | diameter of grains [m] |
| $\rho_s$ | density of dried sediment [kgm$^{-3}$] |
| $\rho_w$ | density of water[kgm$^{-3}$] |
| $g$ | gravity acceleration [ms$^{-2}$] |


**Data Availability**
The modelling dataset including the simulations produced for the present study covers a volume wider than 2 TB. Such
an amount of data raises an evident problem in order to make them available on data repositories. Consequently, the
output of the simulations won't be directly available. However, the model set-up and all the files necessary for their
reproduction will be made available in MIKE FM format upon request to the corresponding author.

**Team list**
Antonio Guarnieri (first and corresponding author), Sina Saremi (co-author), Andrea Pedroncini (co-author), Jakob H.
Jensen (co-author), Silvia Torretta (co-author), Caterina Vincenzi (co-author), Marco Vaccari (co-author).

**Author contributions:**





Antonio Guarnieri implemented the numerical models and simulations, post-processed the raw output, analysed the
results and wrote the manuscript;
Sina Saremi gave technical and scientific support during the implementation of the models, provided the code for the
propellers modelization as input to MIKE and supported the writing and finalization of the manuscript;
Andrea Pedroncini first conceived the idea of the methodology adopted in the study, gave scientific support for the
implementation of the models and feedback during the writing of the manuscript;
Jacob H. Jensen provided scientific support and advice regarding the driving mechanisms of naval induced sediment
dynamics;
Silvia Torretta provided technical support for the model implementation and for the observed bathymetry analysis and
reconstruction;
Caterina Vincenzi and Marco Vaccari provided bathymetry data, sediment data and information on dredging activities
and general sediment related issues. They also favored the acquisition of the naval traffic data.

**Competing interests:**
Caterina Vincenzi and Marco Vaccari are employees of the Port Authority of Genova (Autorità di Sistema Portuale del
Mar Ligure Occidentale), which commissioned and funded the present study to DHI, a private not-for-profit
consultancy and research company in the field of water. Andrea Pedroncini, Silvia Torretta, Sina Saremi and Jakob H.
Jensen are DHI employees. Antonio Guarnieri was DHI employee when the study was conducted; he is now employed
at Istituto Nazionale di Geofisica e Vulcanologua (INGV).

**Acknowledgments**
We are grateful to Stazioni Marittime SpA for providing the daily traffic data of the Port of Genoa which was the
starting point for this study. We are particularly grateful to Captain Calcagno of Stazioni Marittime SpA for the
qualified and experienced information he gave on the sediment and vessels' dynamics in the port, which helped set up
the numerical models, interpret and rely on the final results.

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
