# Peer review of "Effects of naval traffic on sediment erosion and accumulation in ports: a new model-based methodology"

_Ocean Science, 2020_

## Referee Comment (RC1) · Anonymous Referee #1 · 19 Oct 2020

The manuscript ' Impact of naval traffic on the sediment transport of the Port of Genoa – a modelling study' addresses the effects of vessel propellers jets on hydrodynamics and sediment transport in a passengers harbour (Port of Genoa), by means of a well-known widely used hydrodynamics and sediment transport model (MIKE). Model results are qualitatively compared with real measurements.

The manuscript presents an interesting methodology that can potentially be used as a science-based port management and decision making tool and be further scaled-up to other locations. However, there are some lacks in the analysis methods as well as in the number of datasets/results shown.

The manuscript can be reconsidered for publication, if major comments are addressed.

[Figure]

The current review is assessed by following the key questions of the OS review-criteria (https://www.ocean-science.net/peer_review/review_criteria.html):

1. Does the paper address relevant scientific questions within the scope of OS? Within the scope of OS Special Issue 'Advances in interdisciplinary studies at multiple scales in the Mediterranean Sea' and the general scope of OS, the manuscript stands for a new methodology, that potentially can be the seed for a new operational system to a science-based harbour management.

2. Does the paper present novel concepts, ideas, tools, or data? As aforementioned, the paper presents a potential new methodology that can be upscaled to an operational tool, but it is still in an early stage of development.

3. Are substantial conclusions reached? Manuscript conclusions are interesting, but further analysis should be addressed (commented in the following points).

4. Are the scientific methods and assumptions valid and clearly outlined? Most of the assumptions are clearly presented. However, some questions have risen regarding some assumptions: - Why did authors considered the specified three layers scheme? Is this scheme characteristic of the study area? Is it supported by previous similar works? - Are there relevant differences on considering a three layers bed versus considering a single layer? - Since it is stated that the method can be potentially used in a daily operational system for harbour management, which is the computational time of simulations?

5. Are the results sufficient to support the interpretations and conclusions? No. In the manuscript it is stated that 24 scenarios have been simulated, however only the results from 2 scenarios are shown. It is highly recommended to show the results of the rest of scenarios in some way (a common way is by using a matrix of plots).

6. Is the description of experiments and calculations sufficiently complete and precise to allow their reproduction by fellow scientists (traceability of results)? Similarly to

previous question, despite methodology is well described, the lack of results of the rest of scenarios will make difficult to reproduce the modelling results.

7. Do the authors give proper credit to related work and clearly indicate their own new/original contribution? Yes, authors do indicate the novelty of the method, however, as aforementioned, it should be further proven.

8. Does the title clearly reflect the contents of the paper? No, title should reflect that the manuscript presents a new method, not directly the physics underlying sediment transport processes forced by propeller jets, since it is not addressed.

9. Does the abstract provide a concise and complete summary? The abstract contains the main points addressed in the manuscript, however it states that 'In the present work we study the erosion and sediment transport induced by...', while a relevant part of the manuscript addresses hydrodynamics. Moreover, more than 'erosion and sediment transport', it stands for erosion/deposition patterns.

10. Is the overall presentation well structured and clear? The manuscript structure is appropriate, however the fluency of the discourse along the sections should be revised (see next point comments).

11. Is the language fluent and precise? - No. Language should be reviewed in depth and be more precise. Along the manuscript, language is redundant and not focused to the point of the results or discussion. Some concepts are repeated within consecutive paragraphs.

- It is advised to use shorter and concrete sentences along the manuscript.

- Furthermore, there are basic mistakes on the formal format on literature citation within the manuscript that must be revised along the whole document. For instance: Line 344: '. . . settling according to Winterwerp (Winterwerp and Van Kesteren, 2004). . .' should be replaced by '. . . settling according Winterwerp and Van Kesteren (2004). . .' Idem at lines 324, 345, 348, 383, and lot more along the manuscript and Appendices.

- Review the sentence in lines 553-555. What is it supporting to the overall discussion and conclusions?

12. Are mathematical formulae, symbols, abbreviations, and units correctly defined and used? As aforementioned, citations of formulae should be properly written.

13. Should any parts of the paper (text, formulae, figures, tables) be clarified, reduced, combined, or eliminated? A high quality scientific paper should be concise in terms of the objectives to be addressed. While the objectives are more or less stated in the abstract, they are not present in the Introduction, where they should appear clearly stated. Results and discussion are together in the Results section. It is recommended to change Results section name.

14. Are the number and quality of references appropriate? Yes.

15. Is the amount and quality of supplementary material appropriate? Yes.

---

## Referee Comment (RC2) · Anonymous Referee #2 · 27 Oct 2020

[referee-annotated manuscript omitted]

---

## Author Comment (AC2) · 6 Nov 2020

[revised manuscript text omitted]

---

## Author Response (AR1)

**Answers to the interactive comment by anonymous referee #1**

The answers to the interactive comments by anonymous referee #1 to the manuscript *"Impact of naval traffic on the sediment transport of the Port of Genoa – a modelling study"* follow. They have been shared with the co-authors of the manuscript. The numbering follows that of the referee's comments.

1. The comment does not require specific answers;
2. The comment does not require specific answers;
3. Answers are given within those to comments 4 to 7;
4. We agree that we introduced the three layer model without a thorough explanation of this choice, probably giving for granted the fact that a three layer bed model is more complex and potentially accurate than a one or two layer model, thus allowing intrinsically to represent the real physical processes in a more realistic way.
   The degree of consolidation of the bottom sediment is time and depth dependent. The surface layer - which directly contributes to the injection of material into the water column - is consequently much less consolidated than the lower layers, since there is no matter above it and since it is composed by freshly deposited sediment due to the continuous rework it is subject to. This is even enhanced in a port environment where the bottom is continuously influenced by the propellers' induced jets acting several times per day. To account for this a multilayer bottom model would be recommended. In fact, a single layer bed representation would imply an overestimation of the bed erodibility (soft mud, thus easily reworked), resulting in unrealistic further overestimations of sediment erosion and concentration along the water column. However, we considered that a bed composed by only two layers would also not be appropriate because it would have not allowed to account for a gradual transition from unconsolidated to consolidated material, causing an unrealistic abrupt passage between erodible and stable bed. This induced us to consider an intermediate layer allowing for a smoother transition. We will argument better these concepts in the revised version of the article.
   For what concerns the computational effort, the time needed for a single hydrodynamic simulation is approximately 8 hours for a parallel 20-core simulation using 2.4 Ghz processors, while the time needed for a single simulation of the sediment transport model is approximately 20 minutes with the same computational configuration. For potential operational purposes the hydrodynamic model could be run once in offline mode since the vessels trajectories to and from the same docks are very similar to each other. Then, for every new passage the sediment transport model could be run again in operational model (the short simulation time allows for it) and the bottom change kept up-to-date constantly, according to the actual vessels' passages;
5. As stated in the manuscript, since the shape of the wet basins is similar for all the simulated docks, also the hydro and sediment transport dynamics is similar for all the simulations, provided that the vessels are performing similar maneuvers (all docking operations are conceptually similar to each other, and so are all the undocking operations). This is the reason why only two docks were chosen for the presentation of the results, albeit particularly representative. However, we agree that the results of the bed evolution can be shown for each simulation providing benefit to the manuscript and reliability to the final results. Thus, for the sake of completeness and in order to guarantee a better traceability of results we agree with the referee comment, and we will produce all the 24 maps of total bed change. Nevertheless, we think that introducing so many images in the manuscript would negatively impact the fluency of the reading, so we propose to add the missing results as supplementary material, or at the most as an additional appendix using a matrix of plots, as suggested;

6. We believe that the action to comment number 5 will fulfill also the requests of the present comment;

7. Same answer as number 6;

8. We agree that the title as is might not fully represent the focus of the paper. We will accordingly change it in the revised version referring to the novel proposed methodology and to the erosion/deposition concept, which is the final objective of the article more than sediment transport in general;

9. We agree that we used the expression sediment transport in a way that might be too large (and maybe not fully proper). The abstract should better reflect that the focus of the article is the reproduction of bed erosion and deposition, functional to an optimized management of the ports albeit relevant space was given to the description and interpretation of hydrodynamics and consequent transport of sediment. In the final version we will change the abstract in order to better reflect these concepts, as suggested by the referee;

10. We will proceed with a deep language revision in order to make it more direct, concise and concrete. Long sentences will be divided into a few shorter ones and redundant concepts will be eliminated;

11. Suggestions on the fluency of the language will be followed, the formal mistakes on citations will be corrected and the overall conclusions will be supported to the greatest extent possible. The sentence in lines 553-555 will be revised;

12. Wrong format of citations of formulae will be corrected;

13. The addressed objectives will be clarified in the abstract and better appointed in the introduction. The "Results" section will be changed into "Results and Discussion", since much discussion is performed here, as the referee appointed;

14. The comment does not require specific answers;

15. The comment does not require specific answers.

**Answers to the interactive comment by anonymous referee #2**

The answers to the interactive comments by anonymous referee #2 to the manuscript *"Impact of naval traffic on the sediment transport of the Port of Genoa – a modelling study"* follow. They have been shared with the co-authors of the manuscript.

- **Comment on Line 42:** we have acknowledged the suggested article, but moved the citation in the beginning of the introduction;
- **Comment on line 72:** we added some considerations on the regularity of the most important lines in Section 4.1, when presenting the naval traffic analysis;
- **Comment on line 84:** information on the dimension and mean depth of the basin were added in section "3.1 – Bathymetry";
- **Comment on Line 89:** we added one sentence regarding the interest of Port Authority to the passenger area only;
- **Comments on lines 101, 135:** a short introductory paragraph to the simplification assumption of constant bathymetry as initial bottom condition was introduced at the end of section "2 – Methods". A comprehensive explanation was additionally given in the section "Results and Discussion";
- **Comment on line 153**: such shallow zones (5.0m-7.5m) are actually present only in the eastern boundary of the port. These are marginal areas for our study, rather far from the focus. Moreover, the hydrodynamic model uses sigma coordinates implemented over 10 equally spaced layers. The resulting layer thickness for 5 meter bathymetry would be 50 cm, which we believe would be acceptable for our purposes. Additionally, during the sensitivity study to the grid resolution we also investigated configurations with 20 vertical layers (see section 4.1.1) and explained the issue of increased vertical levels versus computational requirements. We didn't think an insertion in the manuscript was needed for this comment.
- **Comment on line 181:** we are not fully sure we understand the comment. We had the names of the vessels from the schedule provided by the Port Managers. From marinetraffic.com we got the information on the length, width, draught and tonnage of the single vessels. Then, for each dock we calculated the mean of these parameters weighted on the number of annual passages. From these mean parameters we calculated the mean propellers diameters (through empirical formulas). We finally associated to each dock the corresponding representative ship (whose characteristics are those given by the weighted means of the real ones, as explained above). However, to make things more simple we have removed the sentence and replaced it as follows : "The vessels' characteristics necessary to the modelling activity (i.e. length, width, tonnage, draught and typical routes within the port) where deducted from the available information on the web."
- No historical data (year 2017) were required since the vessels' names for the period of interest were given by the Port Managers.
- **Comment on line 197:** no actions needed;
- **Comment on line 250: y**es, This is what we have done in this study. It is not very straight forward since for each time-step of the model we have to define the position of the propellers, and since the propeller is represented through approximately 30 sources and 30 sinks (see description below). We have done this through an ad-hoc offline Matlab code which automatizes the creation of the model set-ups accounting for the positions of the propellers at any time-step. The images of Figure 8 are an example of propeller in different positions at different instants. We believe no action is needed for this comment.

- **Comment on line 268:** brief clarification added in the text;
- **Comment on line 308:** no action needed;
- **Comment to line 313:** clarification on the confinement of the jet was added as well as the citation proposed;
- **Comment on Figure 8:** no actions needed;
- **Comment on line 464:** comment acknowledged; the caption was rewritten;
- **Comment on figure 10:** we increased the size of the label axis and clarified the legend;
- **Comment to figure 11:** we acknowledged the comment;
- **Comment to line 487:** acknowledging the referee's comment and suggestion we have added a comprehensive explanatory paragraph on this issue in the section "Results and Discussion" (from line552 to line 570);
- **Comment to line 517**: yes, it is correct indeed. As explained in the additional paragraph (see comment above) of section "Results and Discussion" we believe our assumption is acceptable and does not compromise the results of the study;
- Comment to line 584: we have added […] "and vessel drafts".

Effects of naval traffic on  sediment  erosion and accumulation in
ports: a new model-based methodology

*Antonio Guarnieri [1], Sina Saremi [2], Andrea Pedroncini[3] Jacob H. Jensen[2], Silvia Torretta[3] Marco Vaccari[4],*
*Caterina Vincenzi [4]*

*(1) Istituto Nazionale di Geofisica e Vulcanologia, Sezione di Bologna, Via D. Creti, 12, 40128 Bologna, Italy*

*(2) DHI, Horsholm, Denmark*

*(3) DHI S.r.l., Via Bombrini 11/12, 16149 Genova*

*(4) Autorità di Sistema Portuale del Mar Ligure Occidentale (Genova), Palazzo San Giorgio - Via della Mercanzia 2*

*Corresponding author: Antonio Guarnieri; antonio.guarnieri@ingv.it*

**Abstract**

The action of  propeller-induced jets on the seabed of ports can cause erosion and the deposition of sediment around the port basin, potentially significantly impacting on the bottom topography over the medium and long time . If such dynamics are constantly repeated for long periods , a drastic reduction in ships' clearance can result through accretion. or it can threaten the stability and duration of  structures through erosion. These sedimentrelated processes  present port managing authorities with problems, both  in terms of navigation safety and in the optimization of  management and maintenance activities of the ports' bottom and infrastructures.

 In this study, which is based on integrated numerical modeling, we examine the hydrodynamics and the related bottom sediment erosion and accumulation patterns induced by the action of  vessel propellers  in the passenger port of Genoa (Italy). The proposed new methodology offers a stateoftheart  science-based tool that can be used to optimize and efficiently plan port management and the seabed maintenance .

**1 - Introduction**

The operational activities of harbors and ports are closely related to the local bathymetry, which must be sufficiently deep  to guarantee the regular passage, maneuvering and berthing of ships. However, ship clearance is often so limited that  the safety of in-port navigation . and ships may even hit the seabed in extreme cases. This is therefore a source of  criticalities that often result in management and maintenance efficiency problems in terms of the bottom and a port's infrastructure in general (Mujal-Colilles et al., 2016; Castells-Sanabra et al., 2020).

The action of a ship's main propellers means that traffic in ports is responsible for generating intense current jets , as noted in Figure 1.Figure 1 The high velocities induce shear stresses on the sea bottom, which can possibly result in sediment resuspension when they exceed the critical stress point for erosion (Van Rijn, 2007, Soulsby et al., 1994; Grant and Madsen, 1979). Before depositing back onto the sea floor the re-suspended sediment may be  transported widely around the basin by the combined effects of natural currents such as those induced by tides, winds or density gradients, and  by vessel-related currents, such as those  induced by the propellers or again by the movement and displacement of the ships. Therefore, theThe continuous traffic in and out ports couldcan thus result in the displacement of a great amounthuge volume of seabed material, which can, in turn, then induce importantsignificant variations ofin the bathymetry in theover medium to long time scales. The result of these variations is the possibleThe formation of erosional or depositional trends forin specific areas of port basins can potentially result from these variations.

[Figure]

**Figure 1 - Example of propeller induced jet of a moving ship (main propulsion without rudder)**

These processes can have direct impact on the operability of ports and on safety depths for navigation (Mujal-Colilles et al., 2016, Castells et al., 2018). If such dynamics are particularly relevantpronounced and fastrapid (bottom accretion of thean order of tens of centimeters per year, or even higher) they induce ), the port authorities tomust undergo dredging operations for the maintenance of the seabed in order, to fully recover the required clearance and operationalensure the conditions necessary for undisturbed ships motion, maneuvering and berthingdocking/undocking operations.

The majorityMost of the published literature and studies about the effects of ships' propellers on port sediments and structures is experimental., and is mainly conducted in laboratories with the use of using physical models (Castells

Mujal-Colilles et al. 2018), while port authorities suffer from the lack of; Yuksel et al. 2019). Few practical instruments are available to for port authorities that can provide robust and scientifically based studiesanalyses and predictions of the describedrelevant processes. Such tools would allow for an aware planning ofcan enable them to plan specific actions aimed at the maintenance ofmaintaining the seabed. This would, and thus help toboth guarantee the continuity in theof operational activities of ports on one side, and tooptimize the optimizationuse of the involved economic resources on the other side. In fact, the need of unplanned. Unplanned maintenance activities usually impliesinvolve additional costs due to operatingthe need to operate in emergency conditions and in some cases to the partial interruption ofpartially interrupt the service.

The integrated numerical modeling of hydrodynamics and sediment transport may representrepresents an important aid to Port Authoritiesport authorities, and more broadly to port managers and operators. It could be used to, as suggested by Mujal-Colilles (2018). This can reproduce and thus provide a better understandunderstanding of the seabed sediment dynamics induced by ships' propellers on theover short, medium and long- time scales and so provide the needed, thus establishing what tools inare required to ensure the perspective of an efficient operational maintenance of the seabed. Such tools can be used in delayed mode in order to reproduce the major sediment processes in the past – as it is the present case – or even in forecast mode through the implementation of real time operational services.

So far, the issue of propeller'sPropeller induced jet has beenjets have mainly been studied through empirical approaches, usually relying either on the German method (MarCorm WG, 2015, Grabe, et al. 2015, Abromeit et al.,

2010,), or on the Dutch method (CIRIA et al., using2007). In such approaches, empirical formulas are introduced in order to estimate the propeller wash on the sea bed in terms of induced velocities and resulting induced shear stresses, dependingbased on specific characteristics of the ships and ports of interest, such as the propeller's bathymetry, propeller typology, diameter, and rotation rate, and ship's draught. The most common approaches are the German method (MarCorm WG, 2015; Grabe, et al., 2015; Abromeit et al., 2010,) and the Dutch method (CIRIA et al., 2007).

The resulting induced velocities are usually only considered only locally for, to inform the technical design of mooring structures and for considerations on the protection of a port's infrastructures in general. Besides the theinfrastructure. Although various assumptions are introduced in thethrough empirical formulas, such an approach is punctualthese approaches are limited and doesdo not provide the full picture offully consider the three- dimensional evolution of the induced jet throughout the water column at any distance from the propeller, inor at any location of the port. The tool isThese tools are therefore not suitable for athe 
[revised manuscript text omitted]

---

## Author Response (AR2)

**Effects of naval traffic on sediment erosion and accumulation in ports: a new model-based methodology**

*Answers to the requests of anonymous referee #1 prior to publication*

The technical issues regarding duplicated table and figure links within the text were corrected. We apologize for the inconvenience which must have occurred when changing format from *.docx* to *.pdf*.

The problems related to references were corrected as well. In particular the citation within the text "Wei-Haur Lam" was supposed to be just "" (Consequently the reference "Wei-Haur Lam" was missing). Additionally, some dates in the references were corrected.

Finally, as requested by the Editorial Support Team, the copyright sign "©" was added in the images of APPENDIX A3 and in Fig.11.

Again, we want to thank both referees for their important critical contributions which helped significantly improve the final version of the manuscript.